# Scaling the Poisson GLM to massive neural datasets through polynomial approximations

**David M. Zoltowski**
Princeton Neuroscience Institute
Princeton University; Princeton, NJ 08544
zoltowski@princeton.edu

**Jonathan W. Pillow**
Princeton Neuroscience Institute & Psychology
Princeton University; Princeton, NJ 08544
pillow@princeton.edu

## Abstract

Recent advances in recording technologies have allowed neuroscientists to record simultaneous spiking activity from hundreds to thousands of neurons in multiple brain regions. Such large-scale recordings pose a major challenge to existing statistical methods for neural data analysis. Here we develop highly scalable approximate inference methods for Poisson generalized linear models (GLMs) that require only a single pass over the data. Our approach relies on a recently proposed method for obtaining approximate sufficient statistics for GLMs using polynomial approximations [7], which we adapt to the Poisson GLM setting. We focus on inference using quadratic approximations to nonlinear terms in the Poisson GLM log-likelihood with Gaussian priors, for which we derive closed-form solutions to the approximate maximum likelihood and MAP estimates, posterior distribution, and marginal likelihood. We introduce an adaptive procedure to select the polynomial approximation interval and show that the resulting method allows for efficient and accurate inference and regularization of high-dimensional parameters. We use the quadratic estimator to fit a fully-coupled Poisson GLM to spike train data recorded from 831 neurons across five regions of the mouse brain for a duration of 41 minutes, binned at 1 ms resolution. Across all neurons, this model is fit to over 2 billion spike count bins and identifies fine-timescale statistical dependencies between neurons within and across cortical and subcortical areas.

## 1 Introduction

The Poisson GLM is a standard model of neural encoding and decoding that has proved useful for characterizing heterogeneity and correlations in neuronal populations [12, 19, 23, 15, 13]. As new large-scale recording technologies such as the Neuropixels probe are generating simultaneous recordings of spiking activity from hundreds or thousands of neurons [8, 21, 4], Poisson GLMs will be a useful tool for investigating encoding and statistical dependencies within and across brain regions. However, the size of these datasets makes inference computationally expensive. For example, it may not be possible to store the design matrix and data in local memory.

In this work, we develop scalable approximate inference methods for Poisson GLMs to analyze such data. Our approach follows from the polynomial approximate sufficient statistics for GLMs framework (PASS-GLM) developed in [7], which allows for inference to be performed with only a single pass over the dataset. This method substantially reduces computation time and storage requirements for inference in Poisson GLMs without sacrificing time resolution, as the sufficient statistics are computed as sums over time.

Our specific contributions are the following. Using quadratic approximations to nonlinear terms in the log-likelihood, we derive the closed-form approximate maximum likelihood and MAP estimates of the parameters in Poisson GLMs with general link functions and Gaussian priors. We introduce

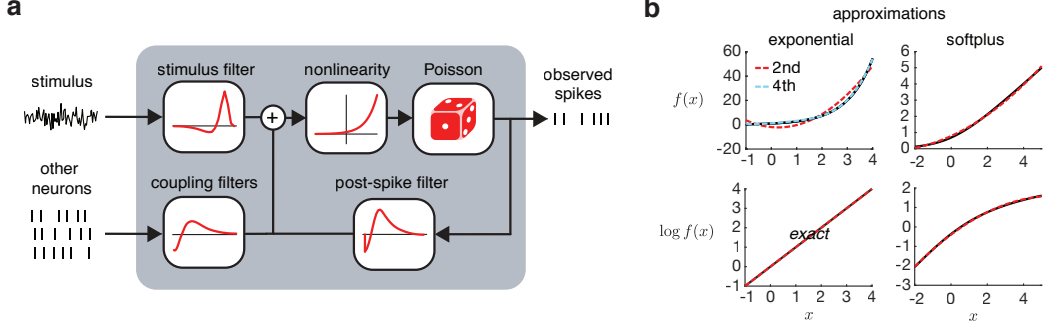

Figure 1: **a.** The Poisson GLM as a model of spiking activity. **b.** Quadratic and 4th-order Chebyshev approximations to $f(x)$ and $\log f(x)$ for $f(x) = \exp(x)$ and $f(x) = \log(1 + \exp(x))$ over two example intervals. The exponential approximation is over an interval for modeling spikes per second and the softplus approximation is over an interval for modeling spikes per bin.

a procedure to adaptively select the interval of the quadratic approximation for each neuron. The quadratic case is the most scalable PASS-GLM because it has the smallest memory footprint, and we found that adaptive interval selection was necessary to realize these benefits. We also show that fourth order approximations are useable for approximating the log-likelihood of Poisson GLMs. Finally, we use the quadratic approximation to derive a fast, closed-form approximation of the marginal likelihood in Poisson GLMs, enabling efficient evidence optimization.

After validating these estimators on simulated spike train data and a spike train recording from a primate retinal ganglion cell, we demonstrate the scalability of these methods by fitting a fully-coupled GLM to the responses of 831 neurons recorded across five different regions of the mouse brain.

## 2 Background

### 2.1 Poisson GLM

The Poisson GLM in neuroscience is used to identify statistical dependencies between observed spiking activity and task-relevant variables such as environmental stimuli and recent spiking activity across neurons (Figure 1a). The model is fit to binned spike counts $y_t$ for $t = 1, ..., T$ with time bin size $\Delta$. The spike counts are conditionally Poisson distributed given a vector of parameters $\mathbf{w}$ and time-dependent vector of covariates $\mathbf{x}_t$

$$y_t | \mathbf{x}_t, \mathbf{w} \sim \text{Poisson}(y_t; f(\mathbf{x}_t^\top \mathbf{w})\Delta). \tag{1}$$

The log-likelihood of $\mathbf{w}$ given the vector of all observed spike counts $\mathbf{y}$ is

$$\log p(\mathbf{y}|\mathbf{X}, \mathbf{w}) = \sum_{t=1}^{T} \log p(y_t|\mathbf{x}_t, \mathbf{w}) = \sum_{t=1}^{T} \left( y_t \log f(\mathbf{x}_t^\top \mathbf{w}) - f(\mathbf{x}_t^\top \mathbf{w})\Delta \right) \tag{2}$$

where we have dropped terms constant in $\mathbf{w}$ and the $t$-th row of the design matrix $\mathbf{X}$ is $\mathbf{x}_t$. The methods in this paper apply both when using the canonical log link function such that the nonlinearity is $f(x) = \exp(x)$ and when using alternative nonlinearities.

### 2.2 Polynomial approximate sufficient statistics for GLMs (PASS-GLM)

With moderate amounts of data, first or second order optimization techniques can be used to quickly find point estimates of the parameters $\mathbf{w}$. However, inference can be prohibitive for large datasets, as each evaluation of the log-likelihood requires passing through the entire design matrix. The authors of [7] recently described a powerful approach to overcome these limitations using polynomial approximations in GLM log-likelihoods. This method leads to approximate sufficient statistics that can be computed in a single pass over the dataset, and therefore is called PASS-GLM. Formally, [7]

considers log-likelihoods that are sums over $K$ functions

$$\log p(y_t|\mathbf{x}_t, \mathbf{w}) = \sum_{k=1}^{K} y_t^{\alpha_k} \phi_{(k)}(y_t^{\beta_k} \mathbf{x}_t^\top \mathbf{w} - a_k y) \tag{3}$$

where the values $\alpha_k, \beta_k, a_k \in \{0, 1\}$ and the functions $\phi_{(k)}$ depend on the specific GLM. In a Poisson GLM with nonlinearity $f(x)$, we have $K = 2$ with $\phi_{(1)}(x) = \log f(x)$, $\phi_{(2)}(x) = f(x)$, $\alpha_1 = 1$, and the other values set to zero. When $f(x) = \exp(x)$, the first function simplifies to $\phi_{(1)}(x) = x$. In [7], each nonlinear $\phi_{(k)}$ is approximated with an $M$-th order polynomial $\phi_{(k)}^M$ using a basis of orthogonal Chebyshev polynomials [11], and the authors show that this approximation leads to a log-likelihood that is simply a sum over monomial terms. They provide theoretical guarantees on the quality of the MAP estimates using this method and on the quality of posterior approximations for specific cases, including the Poisson GLM with an exponential nonlinearity. We use this approach to extend inference in Poisson GLMs to massive neural datasets.

## 2.3 Computing Chebyshev polynomial approximations

We use systems of orthogonal Chebyshev polynomials to compute polynomial approximations. Here, we describe our procedure for computing the coefficients of an $M$-th order polynomial approximation to a function $f(x)$ over the interval $[x_0, x_1]$. We assume that $f(x)$ has a Chebyshev expansion over $[x_0, x_1]$ given by $f(x) = \sum_{m=0}^{\infty} c_m T_m$, where $c_m$ are coefficients and $T_m$ is the degree-$m$ Chebyshev polynomial of the first kind over $[x_0, x_1]$ [11]. By truncating this expansion at the desired order $M$ and collecting terms, we obtain an approximation $f(x) \approx \sum_{m=0}^{M} a_m x^m$. We note that the coefficients $a_m$ for $m = 0, ..., M$ can be estimated by solving a weighted least-squares problem, by minimizing the squared error over a grid of points on $[x_0, x_1]$ between $f(x)$ and an approximation $\hat{f}(x)$ with monomial basis functions. The weighting function is $w(x) = \frac{1}{\sqrt{1-x^2}}$ over $[-1, 1]$ and is mapped to general intervals $[x_0, x_1]$ via a change of variables.

## 2.4 Related work

An alternative approach for efficient inference in Poisson GLMs is the expected log-likelihood approximation [14, 17], which replaces the nonlinear exponential term in the log-likelihood with its expectation across data points. This is justified using knowledge of the covariance structure of the stimulus or by arguments invoking the central limit theorem. The benefits of the polynomial approximation approach are that it applies to arbitrary stimulus and covariate distributions, it does not inherently require large amounts of data, and it can tradeoff storage costs with higher order approximations for increased accuracy. Importantly, in contrast to the expected log-likelihood approximation, the approach in this paper is easily extended to non-canonical link functions.

# 3 Polynomial approximations for Poisson GLMs

## 3.1 Quadratic approximation to exponential nonlinearity

We first apply the polynomial approximation framework to Poisson GLMs with an exponential nonlinearity using a quadratic approximation. With the canonical link function, the $\log f(x)$ term in the log-likelihood is linear in the parameters and we only need to approximate the nonlinear term $f(x) = \exp(x)\Delta$. We approximate this term as

$$\exp(x)\Delta \approx a_2 x^2 + a_1 x + a_0 \tag{4}$$

where the coefficients $a_2$, $a_1$, and $a_0$ are computed using a Chebyshev polynomial approximation over the interval $[x_0, x_1]$ using the methods described in Section 2.3. We currently consider $[x_0, x_1]$ to be a fixed approximation interval and in Section 4.1 we discuss selection of this interval. Example

approximations are shown in Figure 1b. We use this approximation to rewrite the log-likelihood as

$$\log p(\mathbf{y}|\mathbf{X}, \mathbf{w}) = \sum_{t=1}^{T} \left( y_t \mathbf{x}_t^\top \mathbf{w} - \exp(\mathbf{x}_t^\top \mathbf{w}) \Delta \right) \tag{5}$$

$$\approx \sum_{t=1}^{T} \left( y_t \mathbf{x}_t^\top \mathbf{w} - a_2 (\mathbf{x}_t^\top \mathbf{w})^2 - a_1 (\mathbf{x}_t^\top \mathbf{w}) - a_0 \right) \tag{6}$$

$$= \mathbf{w}^\top \mathbf{X}^\top (\mathbf{y} - \boldsymbol{a_1}) - \mathbf{w}^\top a_2 \mathbf{X}^\top \mathbf{X} \mathbf{w} \tag{7}$$

where $\boldsymbol{a_1}$ is a vector with each element equal to $a_1$ and throughout we have dropped terms that do not depend on $\mathbf{w}$. This form has approximate sufficient statistics $\sum_{t=1}^{T} \mathbf{x}_t$, $\sum_{t=1}^{T} y_t \mathbf{x}_t$, and $\sum_{t=1}^{T} \mathbf{x}_t \mathbf{x}_t^\top$.

The approximate log-likelihood is a quadratic function in $\mathbf{w}$ and therefore is amenable to analytic inference. First, the closed-form maximum likelihood (ML) estimate of the parameters is

$$\hat{\mathbf{w}}_{mle-pa2} = (2a_2 \mathbf{X}^\top \mathbf{X})^{-1} \mathbf{X}^\top (\mathbf{y} - \boldsymbol{a_1}). \tag{8}$$

Next, with a zero-mean Gaussian prior on $\mathbf{w}$ with covariance $\mathbf{C}$ such that $\mathbf{w} \sim \mathcal{N}(\mathbf{0}, \mathbf{C})$, the approximate MAP estimate and posterior distribution are

$$\hat{\mathbf{w}}_{map-pa2} = (2a_2 \mathbf{X}^\top \mathbf{X} + \mathbf{C}^{-1})^{-1} \mathbf{X}^\top (\mathbf{y} - \boldsymbol{a_1}) \tag{9}$$

$$p(\mathbf{w}|\mathbf{X}, \mathbf{y}, \mathbf{C}) \approx \mathcal{N}(\mathbf{w}; \boldsymbol{\Sigma} \mathbf{X}^\top (\mathbf{y} - \boldsymbol{a_1}), \boldsymbol{\Sigma}) \tag{10}$$

where $\boldsymbol{\Sigma} = (2a_2 \mathbf{X}^\top \mathbf{X} + \mathbf{C}^{-1})^{-1}$ is the approximate posterior covariance. This enables efficient usage of a host of Bayesian regularization techniques. In our experiments, we implement ridge regression with $\mathbf{C}^{-1} = \lambda I$, Bayesian smoothing with $\mathbf{C}^{-1} = \lambda L$ where $L$ is the discrete Laplacian operator [16, 14], and automatic relevance determination (ARD) with $\mathbf{C}_{ii}^{-1} = \lambda_i$ [10, 22, 18, 25]. In section (4.2), we introduce a fast approximate evidence optimization scheme for the Poisson GLM to optimize parameters in these priors.

## 3.2 Extension to non-canonical link functions

Nonlinearities such as the softplus function $f(x) = \log(1 + \exp(x))$ are often used in Poisson GLMs. We extend the above methods to general nonlinearities $f(x)$ by approximating both terms involving $f(\mathbf{x}_t^\top \mathbf{w})$ in the log-likelihood

$$f(\mathbf{x}_t^\top \mathbf{w}) \Delta \approx a_2 (\mathbf{x}_t^\top \mathbf{w})^2 + a_1 \mathbf{x}_t^\top \mathbf{w} + a_0 \tag{11}$$

$$\log f(\mathbf{x}_t^\top \mathbf{w}) \approx b_2 (\mathbf{x}_t^\top \mathbf{w})^2 + b_1 \mathbf{x}_t^\top \mathbf{w} + b_0. \tag{12}$$

Both sets of coefficients are computed using Chebyshev polynomials over the same interval $[x_0, x_1]$. The approximate log-likelihood is

$$\log p(\mathbf{y}|\mathbf{X}, \mathbf{w}) \approx \sum_{t=1}^{T} y_t (b_2 (\mathbf{x}_t^\top \mathbf{w})^2 + b_1 \mathbf{x}_t^\top \mathbf{w}) - (a_2 (\mathbf{x}_t^\top \mathbf{w})^2 + a_1 \mathbf{x}_t^\top \mathbf{w}) \tag{13}$$

$$= \mathbf{w}^\top \mathbf{X}^\top (b_1 \mathbf{y} - \boldsymbol{a_1}) - \mathbf{w}^\top (a_2 \mathbf{X}^\top \mathbf{X} - b_2 \mathbf{X}^\top \operatorname{diag}(\mathbf{y}) \mathbf{X}) \mathbf{w}. \tag{14}$$

With non-canonical link functions we have one additional approximate sufficient statistic $\sum_{t=1}^{T} y_t \mathbf{x}_t \mathbf{x}_t^\top$. As in the previous section, we can solve this equation to get closed form approximations for the maximum likelihood and MAP estimates of $\mathbf{w}$ and posterior over $\mathbf{w}$. In particular, with a $\mathcal{N}(\mathbf{0}, \mathbf{C})$ prior on $\mathbf{w}$ the MAP estimate (and posterior mean) is $\hat{\mathbf{w}}_{map-pa2} = \boldsymbol{\Sigma} \mathbf{X}^\top (b_1 \mathbf{y} - \boldsymbol{a_1})$ where $\boldsymbol{\Sigma} = (2a_2 \mathbf{X}^\top \mathbf{X} - 2b_2 \mathbf{X}^\top \operatorname{diag}(\mathbf{y}) \mathbf{X} + \mathbf{C}^{-1})^{-1}$ is the posterior covariance.

## 3.3 Higher order approximations

The approximation accuracy increases with the order of the polynomial approximation (Figure 1b). Here, we investigate higher order approximations and return to the exponential nonlinearity. Unfortunately, a third order approximation of the exponential over intervals of interest leads to a negative leading coefficient, and therefore makes optimization of the log-likelihood trivial by increasing the inner product $\mathbf{x}^\top \mathbf{w}$ to infinity. However, a fourth order approximation is useable, which is in contrast

with logistic regression, for which a sixth order approximation was the next useable order [7]. We approximate the exponential using a fourth order polynomial over $[x_0, x_1]$

$$\exp(x)\Delta \approx a_4 x^4 + a_3 x^3 + a_2 x^2 + a_1 x + a_0 \tag{15}$$

using Chebyshev polynomials and compute the approximate log-likelihood

$$\log p(\mathbf{y}|\mathbf{X}, \mathbf{w}) \approx \sum_{t=1}^{T} \left( y_t \mathbf{x}_t^\top \mathbf{w} - a_4 (\mathbf{x}_t^\top \mathbf{w})^4 - a_3 (\mathbf{x}_t^\top \mathbf{w})^3 - a_2 (\mathbf{x}_t^\top \mathbf{w})^2 - a_1 (\mathbf{x}_t^\top \mathbf{w}) \right) \tag{16}$$

$$= \mathbf{w}^\top \mathbf{X}^\top (\mathbf{y} - \mathbf{a}_1) - \mathbf{w}^\top a_2 \mathbf{X}^\top \mathbf{X} \mathbf{w}$$
$$- a_3 \boldsymbol{\mathcal{X}}_3 \bar{\times}_1 \mathbf{w} \bar{\times}_2 \mathbf{w} \bar{\times}_3 \mathbf{w} - a_4 \boldsymbol{\mathcal{X}}_4 \bar{\times}_1 \mathbf{w} \bar{\times}_2 \mathbf{w} \bar{\times}_3 \mathbf{w} \bar{\times}_4 \mathbf{w} \tag{17}$$

where $\bar{\times}_n$ is the tensor $n$-mode vector product and $\boldsymbol{\mathcal{X}}_3 = \sum_{t=1}^{T} \mathbf{x}_t \circ \mathbf{x}_t \circ \mathbf{x}_t$ and $\boldsymbol{\mathcal{X}}_4 = \sum_{t=1}^{T} \mathbf{x}_t \circ \mathbf{x}_t \circ \mathbf{x}_t \circ \mathbf{x}_t$ are the third and fourth order moment tensors summed across data points. The approximate sufficient statistics are the third and fourth order moment tensors in addition to those from the quadratic approximation. We note that computing and storing these higher order moments is expensive and we can no longer analytically compute the maximum likelihood and MAP solutions. Once the approximate sufficient statistics are computed, point estimates of the parameters can be identified via optimization of the paGLM-4 objective.

## 4 Optimizing hyperparameters

### 4.1 Approximation interval selection

When using quadratic approximations for Poisson GLMs we found that the parameter estimates were sensitive to the approximation interval $[x_0, x_1]$, especially when using the exponential nonlinearity (Figure 3a,b). Further, different approximation intervals will be appropriate for different nonlinearities, bin sizes, and neurons (Figure 3). We therefore found it crucial to adaptively select the approximation interval for each neuron and we provide a procedure to accomplish this.

We first generated a set of putative approximation intervals based on the nonlinearity and bin size, which determine the expected centers and lengths of the approximation intervals. For example, with $f(x) = \exp(x)\Delta$ the output of $\exp(x)$ is a rate in spikes per second, so the approximation intervals should be in the range $x = -4$ to $x = 6$, depending on the response properties of the neuron. We found that approximation intervals with lengths 4 through 8 provided a balance between approximation accuracy and coverage of a desired input range for the exponential nonlinearity, while wider intervals could be used for the softplus nonlinearity.

For each interval in this set, we computed the approximate ML or MAP estimate of the parameters. We then computed the exact log-likelihood of the estimate given a random subset of training data, whose size was small enough to store in memory. We selected the approximation interval that maximized the log-likelihood of the random subset of data. We emphasize that different approximation intervals can be efficiently tested in the quadratic case as this only requires solving a least-squares problem for each approximation interval, and the subset of data can be stored during the single pass through the dataset. We note that cross-validation could also be used to select the approximation interval.

Empirically, this procedure provided large improvements in accuracy of the parameter estimates and in the log-likelihood of training and held-out data (Figure 3a,b and Figure 5b). In general, we conjecture that procedures to adapt the approximation interval will be useful for other implementations of PASS-GLMs, for adapting the approximation interval to different datasets and models and for refining the approximation interval post-hoc if the algorithm is making poor predictions in practice.

### 4.2 Marginal likelihood approximation

We are often interested in fitting Poisson GLMs with high-dimensional parameters, correlated input (e.g. arising from naturalistic stimuli), and/or sparse spiking observations. For these reasons, regularization of the parameters is important even when the number of spike count observations is large [18, 5, 20, 6, 14, 3, 9, 1]. In this section, we derive an approximation to the marginal likelihood in Poisson GLMs that follows directly from approximating the log-likelihood with a quadratic polynomial. The approximation is closed-form such that approximate evidence optimization can be performed efficiently, and we use it to optimize ridge and ARD hyperparameters.

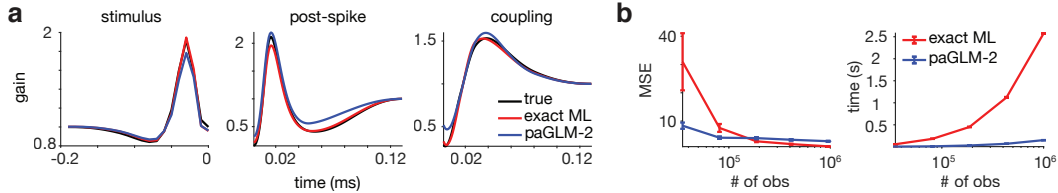

Figure 2: Simulated data experiment. **a.** The exact ML and paGLM-2 filter estimates are similar to the true filters. **b.** The paGLM-2 estimate provides comparable mean squared error (MSE) to the exact ML estimate (left) while the paGLM-2 method shows favorable computational scaling (right).

We restrict ourselves to the exponential nonlinearity, although the same approach can be used for alternative nonlinearities. With a zero-mean Gaussian prior on $\mathbf{w}$ and a quadratic approximate log-likelihood as in section (3.1), we recognize that we can analytically marginalize $\mathbf{w}$ from the joint $p(\mathbf{y}, \mathbf{w}|\mathbf{X}, \mathbf{C})$ to obtain the following approximation to the marginal likelihood

$$\log p(\mathbf{y}|\mathbf{X}, \mathbf{C}) \approx \frac{1}{2}\log|\mathbf{\Sigma}| - \frac{1}{2}\log|\mathbf{C}| + \frac{1}{2}(\mathbf{y} - \boldsymbol{a_1})^\top \mathbf{X}\mathbf{\Sigma}\mathbf{X}^\top(\mathbf{y} - \boldsymbol{a_1}) \tag{18}$$

where $\mathbf{\Sigma} = (2a_2\mathbf{X}^\top\mathbf{X} + \mathbf{C}^{-1})^{-1}$ is the covariance of the approximate posterior and we have dropped terms that do not depend on $\mathbf{C}$. It is important to note that evidence optimization for Poisson GLMs already requires approximations such as the Laplace approximation and our approach is a computationally cheaper alternative, as it does not require a potentially expensive numerical optimization to find the MAP estimate. Instead, the approximate marginal likelihood can be directly optimized using standard techniques.

## 5 Experiments

Throughout our experiments, we refer to estimates obtained using quadratic approximations by *paGLM-2*, estimates obtained using fourth order approximations by *paGLM-4*, and estimates obtained through optimization of the exact log-likelihood by *exact*.

### 5.1 Simulated data

We first tested the approximate maximum likelihood estimates when using a quadratic approximation to the exponential function on simulated data. We simulated spike count data from a Poisson GLM with stimulus, post-spike, and coupling filters when stimulated with a binary stimulus. The stimulus filter had 10 weights governing 10 basis functions, the post-spike and coupling filters each had 5 weights governing 5 basis functions, and a bias parameter was included. In a sample dataset, we found that the exact ML and paGLM-2 estimates of the filters were similar and close to the true filters (Figure 2a, 1 million spike count observations). Next, we compared the scaling properties of the paGLM-2 and exact ML approaches across 25 simulated data sets at each of 5 different amounts of training data. We found that the mean squared error between the true weights and the estimated weights decreased as the number of observations increased for both estimators. The optimization time scaled more efficiently for paGLM-2 than the exact method (Figure 2b, quasi-Newton optimization for exact vs. solving least squares equation for paGLM-2). We used an approximation interval of $[0, 3]$ for each run of this simulation. Interestingly, for smaller amounts of data, the exact ML estimate had an increased mean squared error between the true and fit parameters, as it sometimes overfit the data. Empirically, the approximation appears to help regularize this overfitting.

### 5.2 Retinal ganglion cell analysis

We next tested the paGLM-2 estimator using spike train data recorded from a single parasol retinal ganglion cell (RGC) in response to a full field binary flicker stimulus binned at 8.66 ms [24]. First, we fit a Poisson GLM with an exponential nonlinearity, a stimulus filter, and a baseline firing rate to the responses in approximately 144,000 spike count bins. The stimulus filter was parameterized by a vector of 25 weights which linearly combine the previous 25 bins of the stimulus at each time point, and we set $\Delta = 8.66$ ms such that the firing rate was in spikes per second. On a grid of

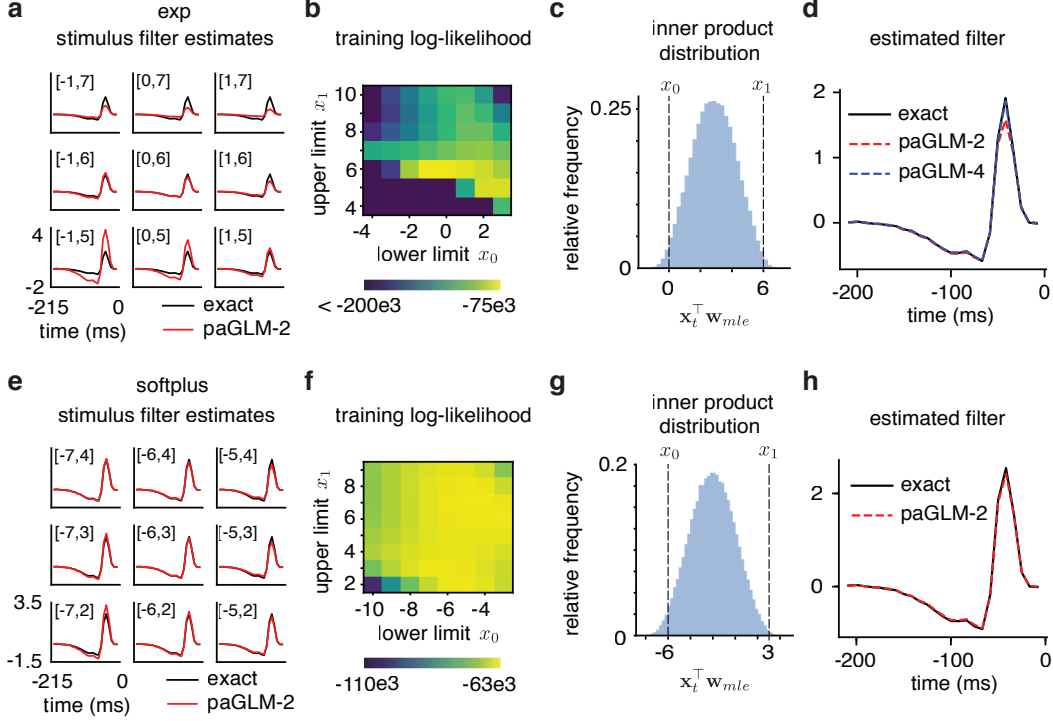

Figure 3: Analysis of paGLM estimators on RGC data with exponential (**a.-d.**) or softplus (**e.-h.**) nonlinearities. **a.** For the exponential nonlinearity with $\Delta = 8.66$ ms, comparison of the exact ML and paGLM-2 estimates of the stimulus filter for different approximation intervals (upper left of each panel). The paGLM-2 estimate can be too large or small relative to the exact ML estimate. **b.** The training log-likelihood (indicated by color) of the paGLM-2 estimate for different approximation intervals. **c.** The distribution of the inner products between the covariates $\mathbf{x}_t$ and the exact ML estimate $\mathbf{w}_{mle}$. The interval $[0, 6]$ covers most of this distribution. **d.** The exact ML, paGLM-2, and paGLM-4 estimates for the exponential nonlinearity computed with approximation interval $[0, 6]$. **e.-h.** Same as (**a.-d.**), except for the softplus nonlinearity and with $\Delta = 1$ such that the rate is in spikes per bin. The approximation interval is $[-6, 3]$.

approximation intervals, we computed the paGLM-2 estimate of the parameters and evaluated the training log-likelihood of the data given the paGLM-2 estimate. The stimulus filter estimates and training log-likelihood varied considerably as a function of the approximation interval (Figure 3a,b). In particular, the paGLM-2 estimate was highly similar to the exact ML estimate for some intervals while too large or small for other intervals. Adaptive interval selection using either the full dataset or a random subset of the data identified the interval $[0, 6]$, demonstrating the importance of this approach. This interval tightly covered the distribution of inner products between the covariates and the exact ML estimate of the weights (Figure 3c). The paGLM-2 and paGLM-4 estimates of the stimulus filter computed over $[0, 6]$ closely matched the exact ML estimate (Figure 3d).

The performance of paGLM-2 was more robust to the approximation interval for the Poisson GLM with a softplus nonlinearity and with $\Delta = 1$, such that the rate is in spikes per bin (Figure 3e,f). Due to the change in nonlinearity and in $\Delta$, the distribution of inner products between the covariates and the exact ML estimate shifted to the left and widened (Figure 3g). The interval $[-6, 3]$ covered most of this distribution, and the paGLM-2 estimate of the stimulus filter computed using this approximation interval was indistinguishable from the exact ML stimulus filter (Figure 3h).

To investigate the quality of paGLM-2 MAP estimates and to verify our procedure for approximate evidence optimization, we increased the dimensionality of the stimulus filter to 100 weights and fit the filter to a smaller set of 21,000 spike counts binned at $\Delta = 1.66$ ms, where the stimulus was upsampled. We used ridge regression to regularize the weights. We selected the approximation interval using a random subset of the data and optimized the ridge penalty by optimizing the approximate log-likelihood (18). The computation time for this procedure, including computing the final paGLM-2

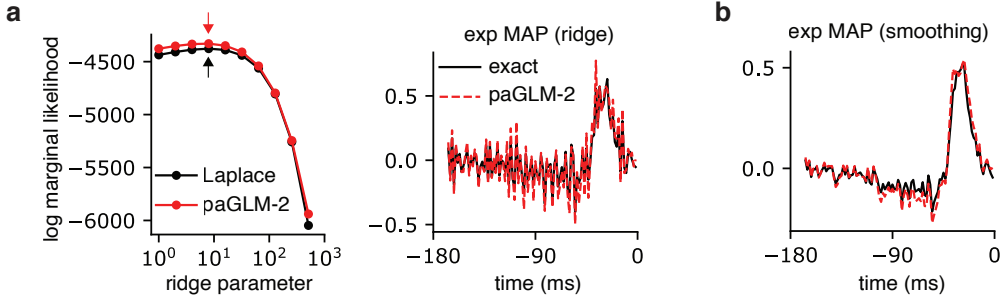

Figure 4: Analysis of paGLM-2 MAP estimators on RGC data binned at 1.66 ms with adaptive interval selection and evidence optimization in a Poisson GLM with exponential nonlinearity. **a.** The approximate log marginal likelihood with a ridge prior computed via the Laplace approximation or via the quadratic approximation (18) (*left*) and the exact and paGLM-2 MAP estimates with the optimal ridge prior value (*right*). **b.** The exact and paGLM-2 MAP estimates with a smoothing prior.

MAP estimate, was 0.7 seconds, compared to 0.7 seconds for computing the exact MAP once and 30 seconds for identifying the optimal ridge parameter using the Laplace approximation. The two methods provided similar estimates of the marginal likelihood and MAP stimulus filter (Figure 4a). Finally, we found that the exact MAP and paGLM-2 MAP estimates computed with a Bayesian smoothing prior were also similar (Figure 4b).

## 5.3 Fully-coupled GLM fit to 831 neurons

We fit a fully-coupled Poisson GLM to the spiking responses of $N = 831$ neurons simultaneously recorded from the mouse thalamus, visual cortex, hippocampus, striatum, and motor cortex using two Neuropixels probes [8]. These responses were recorded during spontaneous activity for 46 minutes. To maintain precise spike-timing information, we binned the data at 1 ms (Figure 5a). For each neuron, the GLM consisted of a baseline rate parameter, a post-spike filter, and coupling filters from all other neurons. We used an exponential nonlinearity such that the firing rate $\lambda_t$ of a neuron at time $t$ was $\lambda_t = \exp(\mu + \sum_{n=1}^{N} h_n * \mathbf{y}_n^{\text{hist}(t)})$ where $\mu$ is the baseline log firing rate and $\mathbf{y}_n^{\text{hist}(t)}$ is the spike train history of the $n$-th neuron at time $t$. We parametrized each filter $h_n$ as a weighted combination of three raised cosine bumps [15]. For each neuron we fit in total 2494 parameters for the baseline rate, post-spike filter, and coupling filters.

To compare the performance of the exact and paGLM-2 MAP estimators on this dataset, we first restricted ourselves to the first 11 minutes of the recording so that we could store the entire dataset and design matrix in memory. We held out the first minute as a validation set and used the next 10 minutes to compute the exact and paGLM-2 MAP estimates with a fixed ridge prior, as hyperparameter optimization was computationally infeasible in the exact MAP case. We used a random subset of the training data to select the approximation interval for each neuron and we computed the exact MAP estimates using 50 iterations of quasi-Newton optimization. We performed this analysis on 50 randomly selected neurons with firing rates above 0.5 Hz due to the cost of computing the exact MAP. On average, the fitting time was about 3 seconds for the paGLM-2 MAP estimates and about 3 minutes for the exact MAP estimates (Figure 5b). Despite the computational difference, the two estimates provided highly similar training and held-out performance. The paGLM-2 MAP estimates computed using the adaptive interval outperformed the estimates using a fixed interval.

We then fit the model to the responses from the first 41 minutes of spiking responses, giving 2.46 million time bins of observations for each neuron. In this case, the design matrix was too large to store in memory (>30 GB). By storing only the approximate summary statistics and one minute of data, we reduced storage costs by a factor of $\approx$40. We computed paGLM-2 MAP estimates using adaptive interval selection and evidence optimization. We placed an ARD prior over each set of 3 coupling weights incoming from other neurons, and optimized the ARD hyperparameters using the fixed-point update equations [2, 18]. We found that sometimes this method under-regularized and therefore we thresholded the prior precisions values from below at $2^6$. Fitting the entire model took about 3.6 seconds per neuron.

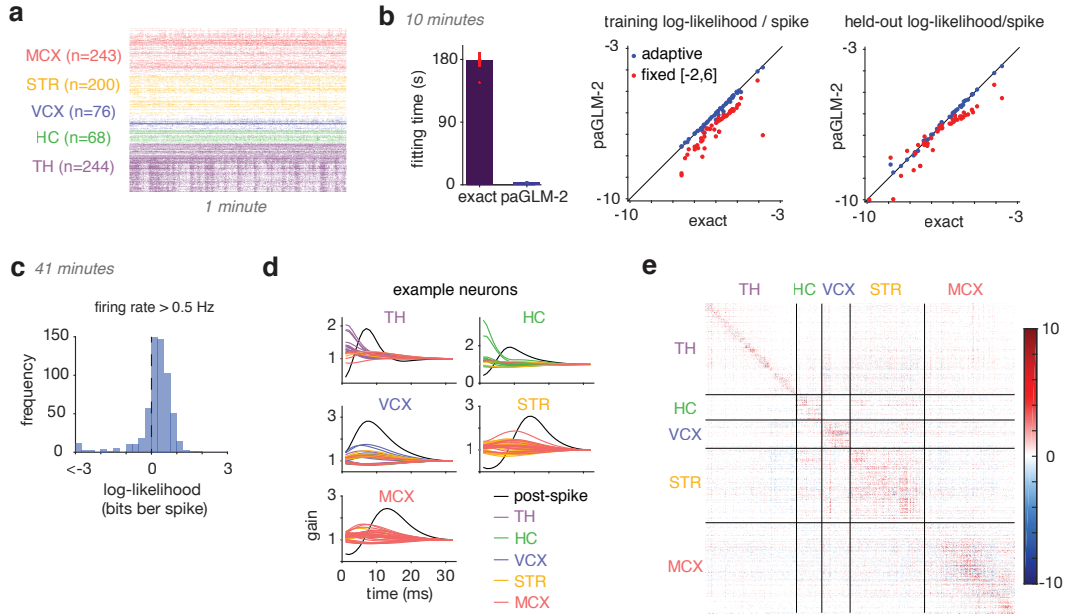

Figure 5: Neuropixels recording analysis. **a.** Raster plot of spikes recorded from motor cortex (MCX), striatum (STR), visual cortex (VCX), hippocampus (HC), and thalamus (TH). **b.** Computation time per neuron (*left*), training performance (*middle*), and held-out performance (*right*) for the exact and paGLM-2 MAP estimates fit to 10 minutes of Neuropixels data, with adaptive and fixed intervals of approximation. The fixed interval $[-2, 6]$ was chosen to tightly cover the optimal approximation intervals across neurons. For **c.-e.**, the model was fit to the first 41 minutes and validated on the last 5 minutes of responses. **c.** Histogram of spike prediction accuracy on validation data for neurons with firing rates greater than 0.5 Hz. **d.** Example paGLM-2 MAP estimates of post-spike and coupling filters. **e.** Coupling matrix with summed MAP estimates of the coupling filters before exponentiation.

The fit model had positive spike prediction accuracy for 79.6% of neurons (469 out of 589) whose firing rates were greater than 0.5 Hz in both the training and validation periods (Figure 5c). This measure quantifies the improvement in prediction we obtain from the fit parameters of the GLM over predictions given the mean firing rate of the held-out spikes. Coupling filters for example neurons are shown in Figure 5d. To summarize the coupling filters across all of the neurons, we computed a coupling matrix whose $i, j$-th entry was the summed coupling filter before exponentiation for neuron $j$ fit to the spiking responses of neuron $i$ (Figure 5e). The rows of this matrix correspond to the set of incoming coupling filters for a neuron. We thresholded values with magnitudes larger than 10 (0.05% of coupling filters) to show the dynamic range. The coupling in the fit model was often stronger within regions and between neurons that were anatomically closer to each other, as the neurons in the coupling matrix are sorted by depth on the probe, with the thalamus, hippocampus, and visual cortex on probe one and the striatum and motor cortex on probe two [8].

## 6 Conclusion

We have developed a method for scalable inference in Poisson GLMs that is suitable for large-scale neural recordings. [1] The method is based on polynomial approximations for approximate sufficient statistics in GLMs [7]. This method substantially reduces storage and computation costs yet retains the ability to model fine time-scale statistical dependencies. While we focused on Gaussian priors in this paper, optimizing the paGLM objective with non-Gaussian, sparsity-inducing priors is an interesting direction of future work. As the approximate sufficient statistics scale with the number of parameters, scaling the method to larger numbers of parameters may require low-rank approximations to the sufficient statistics. Finally, efficient computation and storage of higher order moments will make the more accurate fourth order methods appealing.

**Acknowledgments**

DMZ was supported by NIH grant T32MH065214 and JWP was supported by grants from the Simons Foundation (SCGB AWD1004351 and AWD543027), the NIH (R01EY017366, R01NS104899) and a U19 NIH-NINDS BRAIN Initiative Award (NS104648-01). The authors thank Nick Steinmetz for sharing the Neuropixels data and thank Rob Kass, Stephen Keeley, Michael Morais, Neil Spencer, Nick Steinmetz, Nicholas Roy, and the anonymous reviewers for providing helpful comments.

## Footnotes

[1]An implementation of paGLM is available at `https://github.com/davidzoltowski/paglm`.

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
