[Reviews · NeurIPS 2018]

Reviewer 1



=== Update after author rebuttal === Thanks to the authors for their rebuttal. I have updated my score. One comment about the fully coupled GLM: since the validation data was taken from a specific portion of the experiment (minutes 45-46), then temporal non-stationarities in the data (e.g. changes in arousal state, adaptation, etc.) will make it harder to generalize. Instead, I would recommend that the authors hold out brief snippets of data throughout the experiment to use for validation and testing. === Summary === This paper develops a method for scalable inference methods for Poisson generalized linear models (GLMs). These are particularly prevalent in the neuroscience literature. The authors use the polynomial approximate sufficient statistics (PASS-GLM) method to perform approximate Bayesian inference in these models. They apply this method to two neuroscience datasets: modeling primate retinal ganglion cell responses to white noise (here, the authors validate the approximate methods using exact inference) as well as a dataset of spontaneous activity recorded across brain regions in the mouse. === Review === -- Clarity -- The paper is clear and well written. In particular, I appreciated the thorough background section, especially the comparison with the MELE. -- Originality -- While the applications in neuroscience are new, much of the methods in this paper are directly taken from the PASS-GLM paper. At times, I found it hard to distinguish what (if any) of the method was new, versus deriving the PASS-GLM method for Poisson GLMs specifically. Note that although the experiments in the PASS-GLM paper focus on logistic regression, that paper makes it clear that the method is appropriate for any model in the GLM family, including Poisson regression. Therefore, it seems like much of the novelty of this paper is in the applications (specifically, the application to the large neuropixels dataset recorded across multiple mouse brain regions). Unfortunately, this occupies a small fraction of the overall paper. -- Significance -- In this paper, the authors state (in the introduction) that their contributions are: (a) deriving the expressions for the MLE, MAP, and approximate posterior of a Poisson GLM using the PASS-GLM method, (b) validate the estimators on simulated data and retinal data, (c) introduce two novel extensions of the method, an analytic marginalization when using quadratic approximations and an adaptive procedure for selecting the interval of the polynomial approximation, and (d) apply these methods to study a large dataset recorded across multiple brain regions. Given that most of the method is derived from the ideas in the PASS-GLM paper, I think (a) and (b) are less significant results. Moreover, it is hard to judge the significance of the extensions to the method (c). For example, the authors did not report compare their results on using the adaptive intervals in section 4.2 to using fixed intervals, instead simply stating that they "find that this procedure provides large improvements" (line 185). Large improvements for which application? Simulated data? Retinal data? Mouse data? If the authors had a figure demonstrating this large improvement on real datasets with the adaptive interval, that would greatly increase the significance of this particular contribution. Finally, while the application to mouse data (d) is certainly novel, it is equally hard to assess the significance of these findings. Namely, I think the authors should explicitly answer the following question: what scientific question would neuroscientists be able to answer with the approximate inference Poisson GLM that they could not answer otherwise? For example, am I unable to fit the GLM to the dataset using current techniques? Will it just take longer? How much longer? I would appreciate if there was a rough sense of how long it would take to do the exact GLM given N neurons and T time bins, compared to the approximate version. This presupposed that I even want to fit a massive GLM. The application to the mouse dataset, to me, suggests that this might not be the best idea, as I was quite surprised at how much the GLM overfit to the dataset. The authors report that the model outperforms a homogenous GLM on only 65% of the neurons. Does this mean that the model is strongly overfitting? Shouldn't the hyperparameter search for the regularization parameters have dealt with this? Where was the 1 minute of held out data taken from (randomly throughout the recording, or at the beginning or end?). What if the authors did not fit coupling parameters between all neurons, instead fitting separate GLMs to each area? In that case, are there significant differences in predictive power? What if the full model was uncoupled (only spike history filters)? Also, I would have appreciated more interpretation of the model filters and coupling strengths presented in Fig 4d and 4c. Do these tell us something we did not already know? Do they look significantly different if I took subsets of neurons or data (such that I could do exact inference)? I can appreciate if the authors' goal was to demonstrate that the approximate GLM works on large datasets, rather than analyze this particular GLM model fit, but it is hard to make sense of the significance of being able to fit a large GLM without more exploration of these details. === General comments === This paper does a good job of taking a method recently published in the machine learning community (PASS-GLM) and shows how it may be relevant for neuroscience applications. What I found most difficult about this work is that it is hard to tell which audience this is best suited for. I does not seem like the work is really about novel extensions to PASS-GLM that would be of interest to the machine learning community. Instead, it seems to target those in the neuroscience community who would be interested in building large scale GLMs of their data. However, for a neuroscientist, interested in applications, I think it is difficult to work through the derivations in sections 3 and 4 without a clear understanding of what the approximate GLM buys you, beyond being able to fit a GLM to a large recording (and it is unclear what that buys you in terms of new scientific understanding). I think the paper could be improved by focusing more on applications to neural datasets, with more rigorous comparisons for the new ideas presented in this paper, as well as clear statements about what the approximate GLM buys you in terms of being able to ask new scientific questions. The derivation of the approximate Poisson GLM (section 3), while valuable for those interested in applying the method, is harder to slog through when it is unclear what the method buys you (speaking as a neuroscientist). Perhaps focusing on applications and gains over previous methods first, and using that to motivate the derivation, would be more worthwhile for the community.

Reviewer 2



The paper introduces a straightforward application of the results of ref 5 to the case of Poisson GLMs for large population of neurons. The idea is to replace the terms involving the nonlinearity exp(w’x) by a quadratic or 4th order approximation. This allows then for analytical computation of the posterior under a Gaussian prior and analytic MAP estimates. Because of that, Bayesian regularization techniques can be easily incorporated in that framework. Overall, this is not the most innovative work, but definitely a worthwhile contribution warranted by new recording techniques. The analysis on artificial and RGC data is nice and shows that the obtained approximations are good, even in the quadratic case. The authors should point out clearly that the analysis in Fig. 3 is just for a single cell. Could the authors add a timing analysis to the analysis of RGC data like for Fig. 2? What I don’t quite understand from Fig. 3 is why the ML estimate in panel b is so smooth, while the MAP estimated in e and f are quite noisy. Is that because of finer binning? For the neuropixel recordings, what was the computation time of the fitting procedure? What would be the largest network that could be fit by classic procedures within that time? As the main selling point is a fast approximation that allows fitting larger networks than before, I would be curious about some more discussion on that point. Is the network at a scale where memory constraints are coming into play? Recently a number of DNNs have been used for fitting neural data and GPU implementations are quite fast – how does the GLM compare here? Figure 1: The e of exact has been flipped in panel b

Reviewer 3



The authors consider the problem of fitting Poisson generalized linear (GLM) models to recordings from very large populations of neurons. They leverage recent work to approximate GLM log-likelihoods as polynomials which allow recordings to be summarized with simple sufficient statistics that can be calculated from data. They then show how priors on the GLM weights can be introduced and discuss a method for optimizing hyperparameters of these priors. While the majority of this work focuses on the log link function, they also discuss how their work can be applied with other link functions. They then demonstrate their method on simulated and real data, concluding by fitting GLMs for a population of 831 parameters. I believe this work addresses an important computational neuroscience problem with an eloquent approximation technique that may likely be useful to others, and I believe this paper would provide a valuable contribution to NIPS. I found the description of the core ideas and methods easy to follow and was impressed with the three demonstrations the authors provided. With that said, there are minor points of clarification that may help readers: 1) Equation 7 seems to be missing the term Ta_o. For optimization purposes, this may not matter but still tripped me up a bit, since it is written as equal to 6. 2) I believe the only quantification of run time was done in section 5.1 when estimating an MLE solution without searching for the best approximation interval. It would be nice to have a sense of run-time when also doing evidence optimization to tune hyperparameters and cross-validation to search for the best approximation interval, as it seems in realistic applied settings, both of these may need to be done. Quantification may be unnecessary but it would be nice for the authors to give a sense of the how much relative computational cost is added by these procedures. 3) It would be helpful if the authors could say something about the possibility of applying this method to datasets with 100,000 or more neurons. While we can’t currently record this number of neurons with electrophysiological methods, this is possible in some animal models with calcium imaging and it seems approaches such as this might be adapted for these datasets. However, it seems the optimizations involved here still involve inverting matrices of dimensions equal to the number of neurons. Given this, do the authors have in mind a way this method could be applied when the size of the matrices collected as sufficient statistics grows very large? This is a minor point that I think (at the author’s discretion) might be interesting to include somewhere in the text. After author's response: After reviewing the responses of the other reviewers and the author's response, I still feel this is a high quality paper, worthy of publication at NIPS. I agree with the concerns about the novelty of the work as an application of PASS-GLM, but I still feel the paper is novel enough and addresses an application that is important and likely to become more important in the years ahead to warrant publication at NIPS. Therefore, I have revised my overall score down slightly to 7 these considerations.